# Security Risk Analysis of LoRaWAN and Future Directions

**Ismail Butun [1,*], Nuno Pereira [2]**  **and Mikael Gidlund [1]**

[1] Information Systems and Technology, Mid Sweden University, 851 70 Sundsvall, Sweden; mikael.gidlund@miun.se

[2] School of Engineering (DEI/ISEP), Polytechnic of Porto (IPP), 4200-072 Porto, Portugal; nap@isep.ipp.pt

[*] Correspondence: ismail.butun@miun.se; Tel.: +46-72-595-7333

**Abstract:** LoRa (along with its upper layers definition—LoRaWAN) is one of the most promising Low Power Wide Area Network (LPWAN) technologies for implementing Internet of Things (IoT)-based applications. Although being a popular technology, several works in the literature have revealed vulnerabilities and risks regarding the security of LoRaWAN v1.0 (the official 1st specification draft). The LoRa-Alliance has built upon these findings and introduced several improvements in the security and architecture of LoRa. The result of these efforts resulted in LoRaWAN v1.1, released on 11 October 2017. This work aims at reviewing and clarifying the security aspects of LoRaWAN v1.1. By following ETSI guidelines, we provide a comprehensive *Security Risk Analysis* of the protocol and discuss several remedies to the security risks described. A threat catalog is presented, along with discussions and analysis in view of the scale, impact, and likelihood of each threat. To the best of the authors' knowledge, this work is one of the first of its kind, by providing a detailed security risk analysis related to the latest version of LoRaWAN. Our analysis highlights important practical threats, such as end-device physical capture, rogue gateway and self-replay, which require particular attention by developers and organizations implementing LoRa networks.

**Keywords:** internet of things; sensor node; LPWAN; attacks; threats; vulnerabilities; IoT; analysis; risk; assessment; low power; LoRa; v1.1

## 1. Introduction

The Internet of Things (IoT) is revolutionizing the IT sector and it is predicted that 20 billion IoT devices will seamlessly connect each other to provide information to 3 billion Internet users by the end of 2020 [1]. Thus, the IoT is expected to have a significant impact on our lives in the near future. A special subset of IoT, Low Power Wide Area Network (LPWAN) is significantly increasing its market share and it is projected to have a market worth of 24.5 Billion USD by year 2021 [2].

The IoT is being adopted by many application areas and the communication technologies behind it are still under significant evolution. There exist various communication methods suitable for IoT devices and one way to categorize them is according to the desired wireless communication range. For example, short-range communication technologies such as Bluetooth, ZigBee, and Z-Wave, have been utilized by resource-constrained IoT networks because of their low-energy consumption [3]. On the other hand, these are handicapped by their short-range signal coverage when the application requires tens of kilometers such as in smart cities. Cellular IoT systems have been employed to cope with this problem; however, these systems require high-capacity power supplies along with high-cost hardware and operational cost. All these led to a gap in the IoT communications where technology was needed to provide low power, low cost, and long-range radio communication. To fill this gap, the Low Power Wide Area Networks (LPWAN) technology emerged. Owing to this technology,

resource-constrained sensors/actuators can transmit signals up to 10 s of kilometers (up to 5 km in urban areas and up to 15 km in suburban) and have continuous operation for many years (up to 8–10 years) without an external power source [4].

Application domains of LPWAN are expected to grow in various fields, including the following example areas but not limited to [2]:

- **Smart Gas and Water Metering:** This field is expected to witness the biggest growth rate. The traditional power grid started to be replaced by its smarter counterpart (smart-grid) and therefore smart-metering is one of the most important components, and which lets us understand energy consumption at all the levels of the power grid. The adoption of LPWAN technology by this sector would promote rapid standardization among various smart-meter manufacturers and vendors.
- **Oil and gas operations:** LPWAN connected devices and systems can provide more efficient oil and gas operations, by requiring minimum human-intervention, and constitute a higher value than legacy technologies.
- **Smart streetlights:** According to [4], a special application of LPWAN called LoRaWAN (this technology will be described thoroughly in the remaining part of this manuscript) has been already selected as an enabling technology to monitor not only light illumination levels, but also traffic intensity and air pollution levels of a city in smart city application.
- **Livestock monitoring:** It has been reported that, in another project (*Cattle Traxx*), yet again LoRaWAN has been used to track cows of a farm. End-devices are attached to the ear tags of the cows and gateways around the farm provide the position of the cows with the command center (attached to the application server) [4]. No GPS sensor is attached to the cattle, which leads to less power consumption, eventually less battery usage and less equipment installation on the animal. Hence, traditional triangulation methods are used to calculate the relative positions of the cows from the gateways. Therefore, three or more gateways are deployed in the field. The positions of the gateways are tracked with GPS sensor.

LoRa is developed by Semtech Inc. [5] as a proprietary physical layer protocol to provide low-power and long-distance communication up to 20 km (unobstructed line-of-sight) by using a special radio modulation technique: Chirp Spread Spectrum (CSS) [5]. LoRaWAN is the higher layer protocol (MAC, network and higher layers) based on the LoRa (physical layer), in which the operation and structure of the whole system are defined in detail. Section 3 covers more details of LoRaWAN thoroughly, including presentations related to the network architecture and key distribution mechanisms.

We believe LoRa by Semtech is superior when compared to other LPWAN competitors such as SigFox (SigFox Inc. [6]), NB-IoT (3GPP [7]), Weightless (Weightless Special Interest Group [8]), WAVIoT (Waviot Inc. [9]), Nwave (Nwave Technologies Inc. [10]), RPMA (Ingenu Inc. [11]), UNB (Telensa Inc. [12]), and Qowisio (Qowisio Inc. [13]), due to following unique features [14]:

- High community support: LoRa users (the implementers and researchers) constitute a big community and they support each other. For example "The Things Network" is a crowd-sourced community from 85 countries building a global, public LoRa-based IoT data network [15].
- Installation cost: LoRa end-devices are cheap when compared to its rivals and this is important for large deployments.
- Openness: LoRa is an attractive option due to the open nature of the upper layers, with its LoRaWAN open specification [16], the availability of off-the-shelf and low-cost platforms (see, for example, the Gumstix Conduit Dev Boards [17]).

The v1.1 of LoRaWAN [16] was released on October 2017 and constitutes the main focus of this paper. The main difference between LoRaWAN v1.0 and v1.1 is that with v1.1 roaming of end-devices

is allowed and made possible with the introduction of an extra server called "Join Server". LoRaWAN v1.1 also introduced several security improvements.

As in the case of any computer system, security is one of the biggest concerns in LoRaWAN. The proliferation of IoT devices (especially LPWAN/LoRaWAN devices) is dependent on the public acceptance of these devices as a part of a trusted system. So, enhancing the security level of LoRaWAN devices is very important to acquire public support and acceptance. Therefore, in this work, the authors' aim is to open an avenue to the researchers and practitioners in the field by providing an analysis of the security of the latest version of LoRaWAN (v1.1).

As mentioned earlier, LoRaWAN is a worldwide-deployed IoT protocol. Several discussions and extensive analysis have been made in the literature on the security of LoRaWAN v1.0 and these are presented in Section 2. In a nutshell, it has been shown that v1.0 suffers from several weaknesses, which may allow adversaries to exploit security breaches in the network. These security breaches impact the network availability, data integrity, and data confidentiality. These attacks do not lean on potential implementation or hardware bugs, being instead based on weaknesses of the protocol. Likewise, they do not entail physical access to the targeted equipment and are independent of the means used to protect secret parameters.

To address the mentioned security problems and to provide new functionality, LoRaWAN-Alliance has introduced a new version of LoRaWAN: v.1.1. In this article, we explore the security threats to LoRaWAN v1.1 and we provide practical recommendations aiming at thwarting possible attacks, while at the same time being compliant with the specification, and keeping the interoperability between patched (updated) and non-patched equipment. We present a threat catalog for LoRaWAN v1.1 and analyze these threats in view of the scale, impact, and likelihood of each threat. Finally, we also discuss aspects that are more related to the practical implementation of the technology and while they are not necessarily part of the specification, they are important to highlight for practitioners.

LoRa can be considered as the WiMAX (IEEE 802.16) of the IoT technology. Several attacks (Distributed Denial of Service and Water Torture attacks) against WiMAX has shown to be dreadful [18]. These attacks not only were degrading the network performance but also were diminishing the battery reserves drastically. The latest version of the WiMAX standard, namely 802.16m-2011, has introduced a theoretically more robust security protocol and included encryption for most of the control messages which in turn is expected to thwart most of the previously recognized attacks. As such, LoRaWAN should better be improved in the next releases to mitigate the security risks presented in this work.

Our aim in this research was shedding light on the security vulnerabilities of LoRaWAN v1.1. While doing so we have used ETSI guidelines and also performed a thorough literature search to include every possible attack scenario against the presented protocol. Not only security weaknesses and vulnerabilities are identified, but also remedies and defense mechanisms are also presented for contributing future development of this technology.

The structure of this article is as follows: Section 2 provides related work presented in the literature. This is followed by Section 3 containing an introduction to LoRaWAN 1.1 along with security changes and related assumptions. Then, in Section 4, newly introduced security related problems and vulnerabilities of LoRaWAN 1.1 are presented in an iterative way. Section 5.1 presents security risk assessment methods defined by ETSI. Section 5.2 summarizes the most imminent threats against LoRaWAN v1.1. In Section 5.3, all the risks resulting from the threats that are provided in the previous section are evaluated for LoRaWAN v1.1 by using the ETSI security risk assessment methodology. Section 6 provides suggestions to improve the security of LoRaWAN v1.1 based on the security concerns listed in Section 4 and also the security risk assessment presented in Section 5.3. Finally, the article ends with Section 7, which is comprised of conclusions and future work.

## 2. Related Work

Antipolis and Girard of Gemalto stressed out a problem with LoRaWAN's key provisioning method [19]. In the LoRaWAN v1.0, the Network Server (*NS*) generates both session keys: the *network*

*session key* to be used by *NS* and the *application session key* to be used by the Application Server (*AS*). As the authors mentioned, this would cause a conflict of interest between the network server and the application server. As a solution to this problem, the authors proposed a new LoRaWAN network structure with a trusted third party such as Public Key Infrastructure (PKI).

In [20], Tomasin et al. describe security problems of LoRaWAN v1.0 regarding the *DevNonce*, used during the Over-The-Air-Activation (OTAA) procedure in which the end-device "joins" the network via *NS* (*join_request* and *join_accept* messages). To prevent replay-attacks during the *Join-Request*, a randomized number (*DevNonce*) is included in the message by the end-device so that the *NS* can control whether the *DevNonce* appended to the message has been used by the same end-device already. However, these nonces are produced by using a random-number-generator with limited capabilities such as a limited pool of numbers (each time a number from this pool is selected randomly, the probability of selecting the same number increases as time advances), which might end up with repeating itself after some certain usage time. Accordingly, the authors deduce that, with some special kind of jamming techniques, these *DevNonce* number pool can be diminished in a short period of time. As a result, after then, all of the *join_request* messages from the end-device would be denied by the *NS*.

The *DevNonce* of LoRaWAN v1.0 has been analyzed by also Zulian [21]. The author has shown mathematically that with the *DevNonce* generation system of LoRaWAN v1.0, the end-devices can be unavailable with a certain probability. Author has proposed that increasing the size of the *DevNonce* field to 24–32 bits would mitigate this problem. Luckily, *DevNonce* generation is no longer needed in LoRAWAN v1.1, because the new mechanism issues counting numbers for the nonces, which are seriously tracked to prevent replay attacks [16].

In the proposed LoRaWAN v1.0 security architecture of Naoui et al. [22], proxy nodes are used, which perform several other functions, including the basic function of the conventional LoRaWAN v1.0 gateway. However, the authors do not provide details of how to incorporate these proxy nodes into the existing LoRaWAN v1.0 architecture without causing any problems in the nominal operation.

Miller [23] has presented LoRaWAN v1.0's possible vulnerabilities and related countermeasures. His findings revealed that the vulnerabilities can occur in the phases of key management, communications, and network connection.

Kim and Song [24] proposed a dual key-based activation scheme to tackle various security problems related to the root key of *AppKey* in LoRaWAN v1.0. One of their proposals is the usage of a second root key, to separate the generations of application and network session keys from the same root key. This proposal is, to a certain extent, implemented in the latest version of LoRaWAN(v1.1).

Na et al. [25] claimed that the join request message sent by the end-device to the NS during the OTAA procedure is not encrypted and vulnerable to replay attacks. They have proposed a remedy to prevent this kind of attack. However, in the LoRaWAN, NS is keeping the list of used DevNonce's and automatically protects the network from the bad implications of the replay attacks.

Disregarding the version being used for LoRaWAN, the network might be vulnerable to intra-network interference [26] as well as the jamming attacks, due to the communications through open air. The potential risk is not as serious as in the other narrow-band wireless technologies. This is because the CSS modulation of the LoRa technology spreads the use of the channels to a wider band. Especially, as discussed in [4], LoRaWAN has the best interference immunity characteristics among other LPWAN solutions (e.g., Sigfox). However, as mentioned in [27], novel jamming attacks, such as selective-jamming, can still be successful against LoRaWAN, and these will affect the availability of the network.

Voigt et al. [28] proposed methods to improve the capacity of a LoRaWAN v1.0 network under inter-network interference situation. According to their results, DER has been improved by 33% using directional antennas, and 133% by using additional gateways. Therefore, if the LoRaWAN network is to be used in a hostile environment where jamming or inter-network interference is a relevant problem, these proposed methods can be used to enhance the availability of the network.

Yang [29] has presented several vulnerabilities of LoRaWAN v1.0. This author proposes what is referred to as "bit-flipping attack", consisting of a "man-in-the-middle attack" in which an attacker modifies the messages in between the NS and the application server. Hence, assuming that the communications between these servers are not secured; LoRaWAN v1.0 is susceptible to this kind of attack. Under the same assumption, the vulnerability is still valid for LoRaWAN v1.1, in which several additional servers might exist.

Surprisingly, Lin et al. [30] proposed the usage of block-chain technology, in order to increase the trust value of LoRaWAN v1.0 technology. They argued that the block-chain can be used among several *NS*'s to verify each data transaction. In LoRaWAN, the main aim of the design is decreasing the cost while maintaining some level of security and trust. Besides, in LoRaWAN (in both versions), all servers are assumed to be trusted, hence there is no verification of the data transactions in between the servers is required. Although this might be a weak point of the architecture, conventional Secret-Key-Cryptography encryption algorithms might be used in between the servers of LoRaWAN in order to make certain that end-to-end security is provided.

Sanchez et al.'s work [31] addressed security issues of LoRaWAN v1.0 and offered some remedies by proposing a key management approach. The proposed approach is based on a variation of the Diffie Hellman approach and declared as a convenient solution due to its flexibility in the session key updates, its less computational cost along with fewer message exchange requirements. However, the proposal does not work with the latest version of LoRaWAN (v1.1). Therefore, their work needs to be expanded and revisited, considering the significant security improvements included in v1.1.

For enhancing the privacy of the users in LoRaWAN v1.0, You et al. [32] proposed a solution to prevent a malicious network server from breaking the end-to-end security between *Application Server* and *End-Device*. Although the proposed protocol might be applicable to v1.0 of LoRaWAN, authors did not consider application and inclusion of new version of LoRaWAN (v1.1) which is completely different from v1.0 in terms of network architecture and also messaging (refer to Section 3 for further details on LoRaWAN v1.1). Therefore, the proposal will be useless for the LoRaWAN v1.1.

Haxhibeqiri et al. surveyed papers related to LoRaWAN technology [33] and provided SWOT (Strength, Weakness, Opportunity, Threat) analysis accordingly. They stressed many strengths and opportunities of this technology along with weaknesses and threats when compared with other IoT enabling technologies. According to their findings, LoRaWAN main strengths are; (1) Offering cheap end devices and possibilities of deploying private networks, (2) Removing the need for service subscription such as the one in SigFox or NB-IoT, (3) Decreasing the operational costs for dense IoT deployments, (4) Having large coverage with single gateway, and finally (5) Providing low power operation for end nodes. LoRaWAN main weaknesses are identified as follows; (1) Security issues (especially for LoRaWAN v1.0, many of them are already addressed with LoRaWAN v1.1), (2) Network scalability (already solved with LoRaWAN v1.1), and (3) Malfunctioning of ADR (Adaptive Data Rate) mechanism in congested traffic.

Donmez et al. [34] focused on the security of LoRaWAN v1.1 in backward compatibility scenarios. According to their findings, handover-roaming enables more possibilities for a Man-In-The-Middle (MITM) attack, as the unprotected *FRMPayload* messages are first transported from the serving-NS to the homing-NS, and from there to the AS.

We have reported some early findings of our research in [35], however this paper presents further detailed security analysis and results along with comprehensive discussions flavored with the future suggestions in improving the security of LoRaWAN for the next releases.

## 3. Introduction to LoRaWAN v1.1

Figure 1 shows the network architecture of LoRaWAN v1.1, which is in line with the specifications. Here, as might be seen, a new server called "Join Server" is introduced to orchestrate the OTAA procedure in a more secure way. Besides, with this latest version, instead of single NS, there are three

NS's introduced: "home", "forwarding" and "serving". The rationale behind this inclusion is to enable roaming of the devices citywide, countrywide or from a higher perspective worldwide.

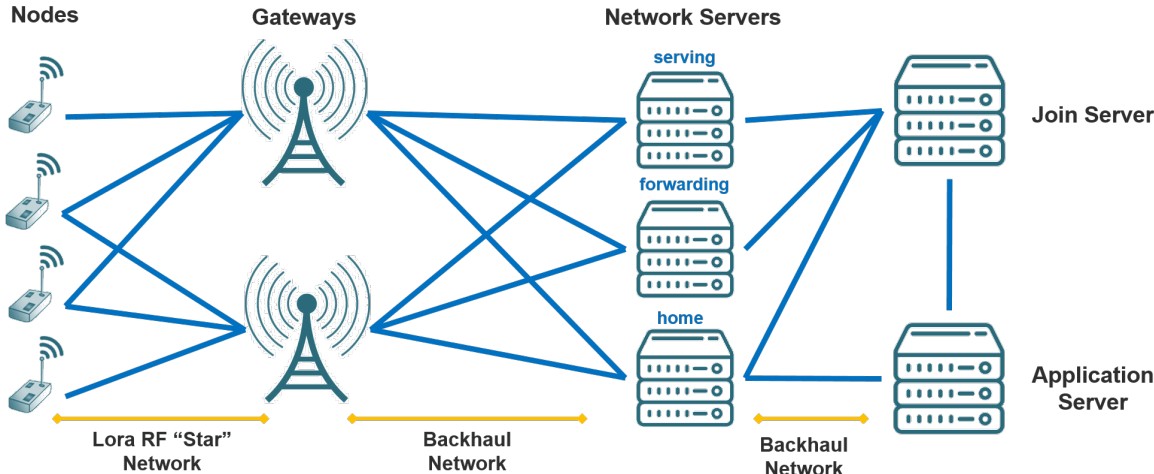

**Figure 1.** LoRaWAN v1.1 network architecture layout.

As in the case of LoRaWAN 1.0, the end-nodes and GW's are connected with "star" topology, and the rest of the network including GW's and servers are connected to each other via "fully-connected mesh" or "partially-connected mesh" topologies (most preferably connected via fiber optics), which is beyond the scope of this manuscript.

Readers who are more interested in the details of how LoRaWAN v1.1 defers from v1.0, may refer to [35]. There, a comprehensive comparison table is provided to enlighten readers about what changes are introduced with the new version, especially from the security point of view (new keys, nounces, frame counters, etc.).

To participate in a LoRaWAN network, each end-device has to be personalized and activated. Activation of an end-device can be achieved in two ways, either via Over-The-Air Activation (OTAA) or via Activation By Personalization (ABP).

*JoinEUI* (new) is the unique identifier of the Join Server and it is embedded into the end-devices during fabrication. This identifier is used during Join procedure. *JoinEUI* is required by only OTAA. *DevEUI* (new) is the unique identifier of the end-devices, and will be labeled (this label should be kept with the device, or otherwise the DevEUI number on it should be saved) on the end-devices during fabrication. In OTAA *DevEUI* is required but for ABP *DevEUI* is recommended.

*AppKey* and *NwkKey* (new) are the AES-128 bit root keys from which the session keys are generated. These root keys are specific to each end-device and therefore they are embedded into the end-devices during fabrication too. The root keys are required during OTAA, and not for ABP.

According to specifications, these root keys (*AppKey* and *NwkKey*) need to be stored in a hardware-secure way, e.g., by using tamper-proof/resistant memory elements such as Secure Elements (SE) or Hardware Security Modules (HSM).

AppSKey (see Figure 2) is the session key used between an end-device and the Application server, generated from the root key of *AppKey*. This key is used to encrypt/decrypt Application layer payloads. *AppSKey* is specific to each end-device. NS's are considered as trusted parties to honestly transmit these encrypted messages to each end. *AppSKey* should be kept secret from outsiders by storing it in a secure way.

From root key of *NwkKey*, several session and lifetime keys are generated (see Figure 2) as described in detail below. Besides, several new counters and parameters are used in v1.1, therefore following subsections are dedicated to an explanation of those.

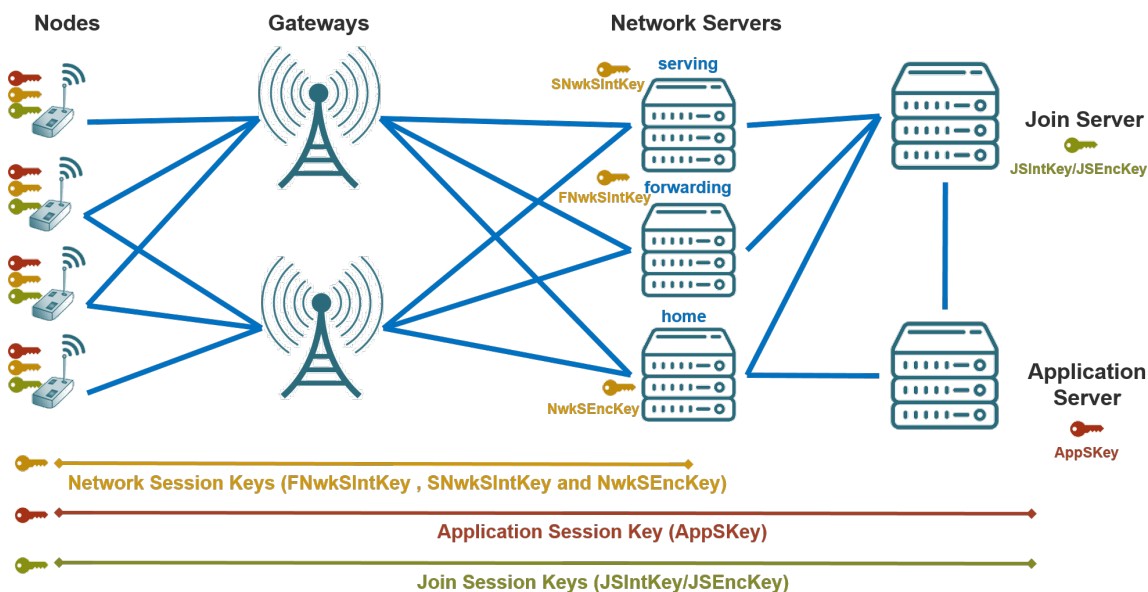

**Figure 2.** An overview of LoRaWAN v1.1 key distribution.

### 3.1. Session Keys

To ensure the integrity of the messages emanating from the network servers, MIC (Message Integrity Code) is appended to all uplink-downlink data messages. Besides, for each end-device, there is a specific network session key. All the session keys are derived by using *NwkKey*, *JoinNonce*, *JoinEUI*, and *DevNonce* value as shown in Equations (1)–(3) (here, note that aes128_encrypt stands for encryption with 128 bit AES Encryption Algorithm). The details are provided below:

(a) Forwarding Network Session Integrity Key (*FNwkSIntKey*—new)

*FNwkSIntKey* shown in Equation (1) is a network session key specific to each end-device that is used for calculating the MIC of all uplink-data messages to ensure data integrity. This key is considered "public" and may be shared with a roaming forwarding-Network server (*fNs*). However, it should be kept secret from outsiders (adversaries) by storing it in a secure way.

$$FNwkSIntKey = aes128\_encrypt(NwkKey, 0x01|JoinNonce|JoinEUI|DevNonce|pad16) \quad (1)$$

(b) Serving Network Session Integrity Key (*SNwkSIntKey*—new)

*SNwkSIntKey* shown in Equation (2) is a network session key specific to each end-device used for calculating the MIC of all downlink-data messages to ensure data integrity. This key is considered "private" and should not be shared with a roaming *fNs*. *SNwkSIntKey* should be kept secret from outsiders by storing it in a secure way (to prevent extraction and re-use by malicious parties).

$$SNwkSIntKey = aes128\_encrypt(NwkKey, 0x03|JoinNonce|JoinEUI|DevNonce|pad16) \quad (2)$$

(c) Network Session Encryption Key (*NwkSEncKey*—new)

*NwkSEncKey* shown in Equation (3) is also a network session key specific to each end-device, and used for encryption/decryption of the MAC layer commands (payload). *NwkSEncKey* also should be kept secret from outsiders by storing it in a secure way.

$$NwkSEncKey = aes128\_encrypt(NwkKey, 0x04|JoinNonce|JoinEUI|DevNonce|pad16) \quad (3)$$

*3.2. Lifetime Join Session Keys*

Join session keys are used for rejoining the network after disconnection(see Figure 2). While generating these keys, besides root key of *NwkKey*, DevEUI is also used. These join session keys are also just required by the OTAA, not ABP:

- JSIntKey (new): Used for the MIC of the *Rejoin-Request* type-1 messages and related Join-Accept answers.
- JSEncKey (new): Used for the encryption of the *Join-Accept* messages in response to *Rejoin-Request* messages.

*3.3. Parameters*

After activation (either by OTAA or ABP), all end-devices should have these items:

- A device address (*DevAddr*—new): Allocated by the *NS* of the end-device.
- Triplet of network session keys: *FNwkSIntKey*, *SNwkSIntKey* and *NwkSEncKey*
- Application session key: *AppSKey*

*3.4. Frame Counters*

Frame counters are incremented as frame traffic is exchanged during the sessions. However, new frame counters are included with this version:

- Network Session: *FCntUp*, *NFCntDwn* (new)
- Application Session: *FCntUp*, *AFCntDwn* (new)

*3.5. New Requirements*

All end-devices (both types OTAA and ABP) now require persistent memory, in order to resume operation. Especially, for ABP, the frame counters will be stored in the persistent memory and there will be no way of resetting the device (going back to factory settings) as in the case of v1.0.

*Join Server* generates the *JoinNonce* value and its provided to the end devices within the join-accept messages each time.

## 4. Security Challenges of LoRaWAN v1.1

In this section, we discuss some security challenges, which might lead to relevant vulnerabilities, in LoRaWAN v1.1. From the general perspective, it can be stated that the newer version of LoRaWAN has benefits of roaming and mobility for the end-devices (by employing extra servers called *JS*, *fNS*, *sNS*) however, this comes at the cost of challenges, problems and security risks described as follows:

*4.1. Network Entities Related*

(a) Gateway related

To our understanding, *GW*s constitute the weakest link of the LoRaWAN. In most of the implementation scenarios, they are deployed few in number (sometimes even 1 or 2). Any kind of capture attack or physical attack would destroy the communication in between the *ED*s and the rest of the network, i.e., servers.

(b) Servers related

Because of roaming, several *NS*'s are required, at least three as shown in Figure 2: *home*, *serving*, and *forwarding*. Besides, one *Join Server* and one *Application Server* are also required. All of these bring the following challenges:

- These several network entities complicate network management. Orchestration of all these *NS*'s and *Join Server*, along with the *Application Server* (see Figure 2 for the architecture of LoRaWAN v1.1) is non-trivial for the service provider.
- Hence all the keys are embedded (installed) upfront to the end-devices; ABP of LoRaWAN v1.1 does not support re-keying of the session keys. In LoRaWAN v1.1, re-keying of the session keys are only possible with *Join-initiation* and *Re-Join* commands during OTAA session. However, there is no apparent mechanism to perform re-keying of the root keys on demand or periodically. A solution needs to be devised so that re-keying would be possible for the root keys as well. A candidate solution can be implementing a mechanism to acquire usage of smart-phones or Bluetooth enabled platforms such as tablets to initiate and perform re-keying of the root keys on-site by using explicit certificates.
- There is a lack of the definition on the session (e.g., an OTAA session) durations of LoRaWAN v1.1: It is ambiguous that how long a session lasts. The dwell time of the signals on air is very well defined in Physical Layer specification, however the session times for the higher protocol layers are not indicated in the specification document.

(c) End-device related

Physical security of end-devices is an important task. To protect the network from physical attacks, especially from device capture attacks, tamper-resistant hardware should be used.

*4.2. Key Distribution Related*

As mentioned earlier in Section 3, there are two mechanisms for distribution of the keys: ABP and OTAA. Both methods have their unique advantages yet also have security implications as detailed below:

(a) ABP related

ABP can be an easy way to simplify deployment, at the cost of reduced security as ABP devices use the same session keys for their lifetimes (i.e., no re-keying is possible). It is important to have a different set of keys per end-device. Counters must be kept in non-volatile memory. If a problem occurs regarding the storage of the counters in the non-volatile memory and also in the process of resuming the final count of the counter following a reset event, then the ED will go out-of-SYNC and be useless after then.

(b) OTAA related

OTAA provides a more flexible and secure way of establishing session keys with the servers. Therefore, it is recommended that OTAA devices are used for higher security demanding applications. However, the following concerns are still valid for OTAA:

- For the OTAA activation procedure, the root keys are needed for generating the session keys. These *NwkKey* and *AppKey* are AES-128 root keys specific to the end-device that are assigned to the end-device during fabrication. These keys need to be securely stored.
- During fabrication *NwkKey/AppKey* key-pair will be generated for each end-device (all related session keys are generated later on by using these specific root keys); but it brings the challenge about keeping the track and taking care of these key pairs.
- PKI can be used for distributing the keys to the end-devices, instead of embedding them during the fabrication process. This would introduce more flexibility and security to the LoRaWAN. Hence, by this way, it would be possible to update the root keys (*NwkKey/AppKey* pair) periodically as well. This remedy is further discussed in Section 6.1 with more details.

*4.3. Implementation Related*

In this section, we discuss potential implementation related vulnerabilities. These vulnerabilities generally occur due to mistakes during application development or installation, when developers fail to follow the specification document closely, due to its complexity or difficulty to put into practice:

(a) Exit procedure related

This is an important procedure to be used for decommissioning of the end-devices after their license ends or they are thought to be compromised and wanted to be cut-off from the network. Here is what is stated in the specification document related to the exit procedure for the end-devices: "The procedure used for deactivating the session is the Exit Procedure, which is the counterpart of the Join Procedure. There is no explicit and dedicated LoRaWAN signaling for performing the Exit Procedure. It is assumed that the End-Device and the backend rely on application-layer signaling to perform this procedure. Triggers and the details of application-layer signaling are outside the scope of this specification." The definition expects exit procedure to the handled by application-layer programming. It is apparent that there is a lack in the standard definition related to the exit-procedure and this might cause complications during the long-term operations. For example, an exit procedure of an end-device should result in with the termination of all IDs, passwords, counters, and nonces related to that specific end-device permanently. Application-layer programmers are strongly advised to be very careful not skipping the inclusion of exit procedures for decommissioning purposes.

(b) DevEUI related

*DevEUI* must be stored in the end-device before the Join procedure is executed during OTAA. *NS*'s hold a list of *DevEUI*'s used throughout the lifetime of the network. Although there exists a procedure to renew the *DevAddr*, the procedure for revoking or renewing *DevEUI*'s is not provided in the specification document. This is somewhat related to the *Exit procedure* described in the previous paragraph and should be included explicitly within the specification for the next release.

(c) Frame counters related

This is a particularly sensitive implementation area both for the end-device and servers. In the past (shown in the literature by Aras et al. [27]), security flaws have risen from mishandling counters. In the specification of LoRaWAN v1.1, the following is stated: "For OTAA devices, Frame counters must not be re-used for a given key, therefore new Session Context must be established well before saturation of a frame counter. It is recommended that session state is maintained across power cycling of an end-device. Failure to do so for OTAA devices means the activation procedure will need to be executed on each power cycling of a device" [16]. The mentioned situation requires saving the frame counters to permanent memory in a timely manner.

(d) JoinEUI related

*JoinEUI* must be stored at the end-devices before starting the OTAA procedure. This means that, any possible change of the Join Server which might required an *JoinEUI* change, would affect all previously loaded end-devices. All of these devices would be useless hence, there is no JoinEUI renewal procedure is defined. In the current version, Join Server is assumed to continue using the same *JoinEUI* for the whole lifetime of the network.

(e) Secure Storage related

In the specifications, it is mentioned that, besides root keys (*NwkKey*/*AppKey* pair), all session keys need to be stored in a "secure-way", so that extraction and re-use of these keys by adversaries would be prevented. It is possible to secure non-volatile memory of the end-devices with the cost of

hardware supported tools such as SE or HSM; however, to our knowledge, the security of volatile memory is not hardware-supported and a challenging task from software implementer point of view.

*4.4. Trust Related*

*NS*'s are considered as trusted parties to honestly transmit encrypted messages to each end. However;

- In the specification of LoRaWAN v1.1, following is stated: "Application payloads are end-to-end encrypted between the end-device and the application server, but they are integrity protected only in a hop-by-hop fashion: one hop between the end-device and the *NS*, and the other hop between the *NS* and the application server" [16]. As mentioned in [24], this means that, a malicious *NS* may be able to alter the content of the data messages in transit, which may even help the *NS* to infer some information about the data by observing the reaction of the application end-points to the altered data. As a conclusion to this statement, it can be stated that end-to-end confidentiality and integrity protection between the servers is not warranted or supported by LoRaWAN v1.1.
- While the specification implements a "federated *NS* infrastructure in which network operators are not able to eavesdrop on application data", there is some trust put on the *NS* in that it will not perform selective forwarding attack or act as a sinkhole.

*4.5. Roaming-Related*

Roaming support is one of the innovative features introduced in LoRaWAN v1.1. However, it brings extra challenges:

- Both versions of LoRaWAN are susceptible to bit-flipping attacks (please refer to next section for details of this attack type) happening in between servers. The inclusion of handover-roaming in v1.1 makes the situation worse.
- As also stressed in [34], handover-roaming enables more possibilities for a MITM attack.
- Handover-roaming can cause a fall-back when the back-end (*sNS*) that serves the roaming *ED* runs an older version of LoRaWAN, i.e., v1.0 [34].

*4.6. Compatibility-Related*

Last but not least, the LoRaWAN is downwards compatible. For the sake of compatibility, the LoRaWAN v1.1 network will able to work downwards compatible with LoRaWAN 1.0 end-devices. This feature is quite interesting in terms of long-term provisioning and marketing of LoRaWAN, but this might introduce additional risks in the future, especially into the network that is consisting of mostly LoRaWAN v1.1 devices.

**5. Security Threat and Risk Assessment of LoRaWAN v1.1**

*5.1. Risk Assessment Methodology*

In this section, we have used the security analysis methodology of Plosz et al. [36], which have been designed by following ETSI guidelines [37]. The methodology includes likelihood and impact assessment of the threats in order to obtain a detailed and categorized security risk analysis as described below:

5.1.1. Likelihood Assessment

Three discrete levels of categorization are provided for this assessment method depending on the likelihood (probability) of a threat: *unlikely*, *possible*, *likely*. Following two factors are used to evaluate the likelihood assessment of threats:

(a) Motivation level

This represents the motivation level of an attacker to perform an attack. Depends on the stakes of the attack result (opportunity) and also related to the desperation degree of the attacker (greed). For example, if the result of an attack would reveal more benefits than the cost of the attack itself, then this would create motivation for the attacker. Motivation level is described by three levels: *low*, *medium*, *high*.

(b) Technical difficulty

This represents the bounds and challenges related to performing an attack. For example, replay attacks are quite easy to perform, requires recording and replaying the same message transmitted by others. Therefore, its technical difficulty is categorized as "solvable". However, introducing an authentic end-device (called "rogue end-device") to the network requires more advanced skills and categorized as "strong". Technical difficulty is described by three levels: *none*, *solvable*, *strong*.

(c) Likelihood of an attack

Based on these risk factors (motivation level and technical difficulty), likelihood assessment chart is created as shown in Figure 3. For example, if an attack is easy to perform (difficulty of none) and if the stakes are good (motivation of high), then the likelihood of the attack is "likely".

| Difficulty / Motivation | None | Solvable | Strong |
|---|---|---|---|
| Low | Unlikely | Unlikely | Unlikely |
| Medium | Likely | Possible | Unlikely |
| High | Likely | Likely | Unlikely |

**Figure 3.** Likelihood of an attack.

5.1.2. Impact Assessment

Three discrete levels of categorization are provided for this assessment method depending on the impact (total result on the victim) of a threat: *minimal*, *moderate*, *significant*. Following two factors are used to evaluate the likelihood assessment of threats:

(a) Scale level

This represents the scale of an affected area because of an attack that is being performed. It can be categorized as three levels: end-device, LoRaWAN, LoRaWAN-EN. "end-device" represents that the attack is only affecting only a single node. "LoRaWAN" represents that the LoRaWAN network is under attack. "LoRaWAN-EN" stands for "LoRaWAN Enterprise Network" and includes multiple LoRaWAN networks.

(b) Detectability and Recoverability

This represents whether the attack is detectable and what is the impact of that, along with the effect of the recovering from that same attack. It can be categorized as three levels: *low*, *medium*, *high*.

(c) Impact of an attack

Based on these risk factors, impact assessment chart is created as shown in Figure 4. For example, if the scale of an attack is network-wide (scale of LoRaWAN) and if the detectability chance is very possible (detectability of high), then the impact of the attack is "moderate".

| Scale \ Detectability | Low | Medium | High |
|:---:|:---:|:---:|:---:|
| End-device | Moderate | Minimal | Minimal |
| LoRaWAN | Significant | Significant | Moderate |
| LoRaWAN-EN | Significant | Significant | Moderate |

**Figure 4.** Impact of an attack.

### 5.1.3. Risk Assessment

According to [37], ETSI categorized and ranked the risk of security threats (by matching the results of Figures 3 and 4) as follows which are also summarized in Figure 5:

(a) Critical Risk

The risk is *critical* if the attack is likely to happen and has significant impact.

(b) Major Risk

The risk is *major*, if the attack is likely to happen and has a moderate impact or, it is possible to happen and has a significant impact or, it is possible to happen and has a moderate impact.

(c) Minor Risk

The risk is *minor* if it is unlikely to happen or has minimal impact.

| Impact \ Likelihood | Unlikely | Possible | Likely |
|:---:|:---:|:---:|:---:|
| Minimal | Minor | Minor | Minor |
| Moderate | Minor | Major | Major |
| Significant | Minor | Major | Critical |

**Figure 5.** Risk assessment guideline.

### 5.2. LoRaWAN v1.1 Threat Catalog

In-line with the security vulnerabilities and concerns provided in Section 4, this section reveals lists the cyber attacks that are possible to be performed against LoRaWAN v1.1.

### 5.2.1. False Join Packets

In implementation related vulnerabilities (refer to Section 4.3), renewal of *JoinEUI* and *DevEUI* values were reported to be problematic. However, even they are old in value, they can still be used for MIC: *Join-Request* and *Join-Accept* messages are protected with MIC (by using *JoinEUI* and *DevEUI* values) and also with another unique nonce (*JoinNonce*) value, it is quite unlikely this attack to happen.

### 5.2.2. MITM Attacks

- **Bit-flipping or Message Forgery Attack:** As also mentioned earlier in Section 2, several security vulnerabilities of LoRaWAN v1.0 are presented in [29]. Especially, a specific version of MITM attack called "bit-flipping attack", in which an adversary (or a rogue *NS*) changes the content of the messages in between *NS* and *AS*. Bit-flipping attack still constitutes a threat for v1.1 as also declared in the specification document [16].

- **Frame Payload Attack:** Section 4.5 described that handover-roaming enables more possibilities for a MITM attack, as the unprotected *FRMPayload*'s are first transported from the *sNS* (serving-NS) to the *hNS* (homing-NS), and from there to the *AS*.

Henceforth, as also discussed in Section 4.4, *NS*s are considered as trusted entities by default. However, deploying engineers are recommended to use extra precautions (end-to-end security solutions) if they are wishing to have end-to-end confidentiality and integrity protection against MITM attacks.

### 5.2.3. Network Flooding Attack

As stressed in Section 4.1, end-devices can be captured and used to perform attacks against the rest of the network. For instance, it is possible for an end-device to degrade the network by flooding it with packets. Nevertheless, end-devices should comply with regulatory airtime restrictions and some networks might protect themselves from flooding attacks by imposing added airtime restrictions.

### 5.2.4. Network Traffic Analysis

This is a passive attack and also called eavesdropping attack if happens in the physical layer (radio signals). In this kind of attack, an attacker can setup a *Rogue-GW* by using the vulnerabilities mentioned in Section 4.1, to receive packets and deduce some knowledge about the data being transmitted or the keying material being used. Observe that, without access to key material, it is infeasible for the attacker to be able to decode the contents of the packets received and the usefulness of this traffic analysis will be highly application specific. For instance, a LoRa network deployed in a smart-city application to deduce noise levels in the city might leak information related to the level of activity in certain locations through simple observation of transmissions.

### 5.2.5. Physical Attacks

Network entity (see Section 4.1) and implementation (see Section 4.3) related vulnerabilities of LoRaWAN can be exploited by the adversaries to perform the following attacks:

- **Destroy, Remove, or Steal End-device:** In both versions of LoRaWAN, root keys (from which session keys are generated) are uniquely generated for each device during fabrication or before deployment. Therefore, revealing of a single root key will not compromise any information in the network other than the data stored at that specific device.
- **Security Parameter Extraction:** The LoRaWAN specification [16] requires protection of the relevant key material against reuse, but nodes need to be adequately protected against firmware change that might indirectly lead to key material reuse.
- **Device Cloning or Firmware Replacement:** An attacker with physical access to the device and can replace firmware or steal/reuse key material. While [16] specifically mentions protection of the relevant key material against reuse, but nodes need to be adequately protected against firmware change that might indirectly lead to key material reuse.

### 5.2.6. Plaintext Key Capture

As mentioned in Section 4.3, if the secure storage elements are not used and keys are stored in regular files (such as text files) in the *ED*'s, then this attack can be a major threat for the confidentiality, integrity and availability of LoRaWAN network.

### 5.2.7. RF Jamming Attack

In the past, it has been shown that by using low cost and commodity hardware jamming of the RF signals is possible [27]. RF jamming attacks in wireless networks lead do Denial-of-Service (DoS), which are generally easy to detect. However, selective RF jamming attacks are harder to detect and

very harmful for wireless communications as it is not trivial to avoid. Therefore, in LoRaWAN, it is possible to jam reception of the signals at a gateway or a node by using RF jamming. This could open-up some advanced attacks as it is described in the Replay attack (refer to Section 5.2.11), which partially relies on the ability to perform selective RF jamming.

### 5.2.8. Rogue End-Device Attack

To be part of the network, an end-device needs access to key materials that should not be accessible to malicious users. One way to do this is by capturing a device and replacing its firmware such that it reuses key materials in the captured device (as discussed in physical-capture-attack, refer Section 5.2.5). End-devices are data sources and are not authorized to reach information at servers. Therefore, any rogue end-device with legitimate authentication information will not able to access or leak data from the network. The most harm can be done by injecting false data to the application server.

End-devices can also be exploited by the attackers to be used as jammers in the network. This might cause the LoRaWAN network to be not available for legitimate end-devices (DoS). However, this will be a restricted area in the vicinity of the Rogue end-device, and the rest of the network continues regular operation.

Rogue end-devices can be also used to perform replay attacks (refer to Section 5.2.11). Packets being transmitted by the neighbors can be captured and replayed later on, which generally will be detectable due to the use of different nonce values (e.g., DevNonce, JoinNonce). However, this might cause waste of available resources in the network and decrease the availability of the *GW* for the legitimate end-devices.

### 5.2.9. Rogue Gateway Attack

From the beginning (including LoRAWAN v1.0), the gateways are always considered as legit and obeying relays. However, this assumption raises the question: "What happens if adversaries hack/capture/replicate one or multiple gateways?" As also mentioned in Section 4.1, *GW*s are vulnerable and therefore can be a target for persistent attackers as described:

- **Beacon Synchronization DoS Attack:** Class B sessions might be vulnerable to Rogue-Gateway attack, which is in line with *Session Hijacking Attacks*. In LoRaWAN, Class B beacons are not secured by any means, indicating that an attacker can set up a *Rogue-GW* to send fake beacons. This could result in class B end-devices to receive messages in windows out-of-sync with the *Rogue-GW* and also increased collisions on packets being transmitted. Addressing this threat would require issuing a key that *GW*'s could use to authenticate beacon transmissions.
- **Impersonation Attack:** Gateways (*GW*s) can also be impersonated to set-up attacks against end-devices (*ED*s). *ED*s can be listened to and their network addresses can be determined. More importantly, a triangulation method (minimum 3 *GW*s are needed in this case to perform the intended capturing attack towards the *ED*) can be used to determine the physical location of the *ED*s.

### 5.2.10. Routing Attacks

Mainly two types of attacks can be considered under this category:

- **Selective Forwarding Attack:** In this attack type, an attacker can selectively forward packets and can cause one or several nodes in the network to be totally blocked, or totally overwhelmed.
- **Sinkhole or Blackhole Attack:** In these attacks, an attacker attracts the traffic through itself by falsely advertising modified routing information. As a result, whole network traffic might collapse.

In LoRaWAN, in order routing attacks to be successful, an attacker needs to capture a *GW* or a server, which is a non-trivial and complicated task. These attacks cannot be perpetrated from the *ED*s, as they do not participate in routing. Therefore, the probability of this attack to happen is very low.

5.2.11. Self-Replay Attack

As discussed in Section 4.2, although OTAA is devised to enhance the security level of the LoRaWAN, itself can be the target of attacks, as details described below: Attacks that are exploiting the join procedure of LoRaWAN v.1.1 is possible by using selective RF jamming attack. An advanced attacker can selectively jam the signals that are being used for OTAA session. In this type of attack, the transmission of a *Join-request* with *DevNonce* from a legitimate *ED* is successfully observed. The corresponding *Join-accept* message from the *NS* to *ED* is jammed by using selective-jamming techniques. After waiting for a timeout to receive the *Join-accept* message from the *NS*, the *ED* retries to join the network and sends again the same *Join-request* message with the same *DevNonce* value as required by the specification. This causes the *NS* to respond the *Join-request*, because still the join procedure is not fulfilled and everything is legitimate and in order according to the specification. This attack will continue till the daily message quota of the *ED* depletes (see Figure 2 for details of the OTAA Join procedure).

This attack is especially valid for LoRaWAN OTAA, since the communication of the messages is limited and under quota. For instance, each *ED* is allowed to transmit at most 14 packets per day (maximum packet payload of 12 Bytes), including the acknowledgments for confirmed up-links [38].

In order this attack to be fully successful, it requires that the attacker can receive packets from the *NS* before the *ED* does and also, at the same time, jam their reception by the *ED*. This can be achieved, for example, by having a device receiving (or sensing) the packet from the *NS* outside the interference range of the jamming device and jam the signal before it reaches the destined *ED*, which might not always be possible depending on the transmission range and distance to the nearest *GW*. This attack is made substantially more difficult for the attacker with the existence of several receive paths for the *NS* packets to the *ED*. Thus, if the *NS*'s transmissions to *ED* can be provided by multiple *GW*s (let us say if the *ED* is under coverage of multiple *GW*s), this attack will be extremely difficult to perform. However, *GW*s are typically deployed fewer in numbers in a LoRa network and therefore this attack has a possibility to succeed.

*5.3. Security Risk Analysis of LoRaWAN v1.1*

Here in this section, we analyze the security threats listed in Section 5.2 with the analysis methods and tools described in Section 5.1 , which was devised by ETSI. Figure 6 depicts the security analysis of LoRaWAN v1.1 by using the methodologies described deeply in Section 5.1. Accordingly, the following results are obtained:

LoRaWAN v1.1 possesses **Minor risk** at security attacks of:

- Bit-flipping or Message Forgery Attack
- Destroy, Remove, or Steal End-device
- False Join Packets
- Frame Payload Attack
- Network Flooding Attack
- Network Traffic Analysis
- RF Jamming Attack
- Selective Forwarding Attack
- Sinkhole or Blackhole Attack

LoRaWAN v1.1 possesses **Major risk** at security attacks of:

- Beacon Synchronization DoS Attack (major risk for availability and minor risk for the rest)
- Impersonation Attack (major risk for availability and minor risk for the rest)
- Plaintext Key Capture (minor risk for availability and major risk for the rest)
- Security Parameter Extraction (minor risk for integrity and availability and major risk for the rest)

LoRaWAN v1.1 possesses **Critical risk** at security attacks of:

- Device Cloning or Firmware Replacement (critical risk for authentication and access control, major risk for confidentiality and integrity and minor risk for availability)
- Self-Replay Attack (critical risk for availability and minor risk for the rest)
- Rogue End-Device Attack (critical risk for authentication and access control availability and minor risk for the rest)

Figure 6 suggests that LoRaWAN v1.1 is more susceptible to physical attacks such as capture, rogue end-device and rogue gateway attacks, rather than attacks towards higher layers of the communications stack, such as network layer attacks (Sinkhole, Blackhole, etc.).

| | Name of threats | Likelihood | | | Impact on | | | | | | Risk at | | | |
|---|---|---|---|---|---|---|---|---|---|---|---|---|---|---|
| | | Technical difficulty | Motivation level | Likelihood | Scale | Detectability | Confidentiality | Integrity | Availability | Authentication & access control | Confidentiality | Integrity | Availability | Authentication & access control |
| | False join packets | None | Low | Unlikely | LoRaWAN | Medium | Minimal | Minimal | Significant | Minimal | Minor | Minor | Minor | Minor |
| MITM Attack | Bit-flipping / message forgery | None | Low | Unlikely | End-device | Low | Minimal | Moderate | Minimal | Minimal | Minor | Minor | Minor | Minor |
| | Frame payload attack | None | Low | Unlikely | End-device | Low | Minimal | Moderate | Minimal | Minimal | Minor | Minor | Minor | Minor |
| | Network flooding attack | None | Low | Unlikely | LoRaWAN | High | Minimal | Minimal | **Significant** | Minimal | Minor | Minor | Minor | Minor |
| | Network traffic analysis | None | Low | Unlikely | LoRaWAN-EN | Low | Moderate | Minimal | Minimal | Minimal | Minor | Minor | Minor | Minor |
| Physical Attacks | Destroy, remove or steal end-device | None | Medium | **Likely** | End-device | Medium | None | None | Moderate | None | None | None | **Major** | None |
| | Device cloning / Firmware replacement | None | High | **Likely** | End-device | Low | Moderate | Moderate | Minimal | **Significant** | **Major** | **Major** | Minor | **Critical** |
| | Security parameter extraction by phy. access | Solvable | Medium | Possible | End-device | Low | Moderate | Minimal | Minimal | **Significant** | **Major** | Minor | Minor | **Major** |
| | Plaintext key capture | None | Medium | **Likely** | End-device | Low | **Significant** | Moderate | Minimal | **Significant** | **Major** | **Major** | Minor | **Major** |
| | RF jamming | None | Low | Unlikely | LoRaWAN | Low | Minimal | Minimal | **Significant** | Minimal | Minor | Minor | Minor | Minor |
| | Self-Replay attack | Solvable | High | **Likely** | End-device | Low | Minimal | Minimal | **Significant** | Minimal | Minor | Minor | **Critical** | Minor |
| | Rogue end-device | Strong | High | **Likely** | End-device | Low | Moderate | Moderate | Moderate | **Significant** | **Major** | **Major** | **Major** | **Critical** |
| Rogue Gateway Attack | Beacon synchronization DoS attack | Solvable | Low | Unlikely | LoRaWAN | Medium | Minimal | Moderate | **Significant** | Minimal | Minor | Minor | **Major** | Minor |
| | Impersonation attack | Solvable | Low | Unlikely | LoRaWAN | Low | Minimal | Minimal | **Significant** | Minimal | Minor | Minor | **Major** | Minor |
| Routing Attacks | Selective forwarding attack | None | Low | Unlikely | LoRaWAN | Low | Minimal | Minimal | **Significant** | Minimal | Minor | Minor | Minor | Minor |
| | Sinkhole / Blackhole attack | None | Low | Unlikely | LoRaWAN | Low | Minimal | Minimal | Significant | Minimal | Minor | Minor | Minor | Minor |

**Figure 6.** Likelihood, impact and risk of attacks against LoRaWAN v1.1.

## 6. Security Suggestions for the Improvement of LoRaWAN

In this section, we provide our suggestions for the further improvement and ramification of LoRaWAN v1.1 as follows:

### 6.1. Usage of Public Certificates for Generation of Root Keys at the End-Devices

Secure provisioning, storage and usage of root keys are very important. A solution to this might be the use of PKI server, such as the one discussed in [19], can be installed to either *JS*, *AS* or *NS*, so that

it can be used as a certificate server. Certificates from the PKI server can be pre-installed during or after the fabrication of the end-devices along with the serial numbers, nonces, and other counters. Root keys can be used by using this certificate and along with an authenticated key-exchange algorithm such as Authenticated Diffie-Hellman key Exchange (ADHE) algorithm [39]. In this scheme, end-device revocation and/or key-update/key-redistribution is an easy task compared to the manual update of the keys (root keys). This topic is very important as it will enable the renewal of the root keys when needed, hence will provide flexibility to the network operator while adding/revoking keys.

*6.2. Nonce Related Improvement*

Each end-device may have nonce which might be associated with the serial number of itself. The *NS* holds the list of nonces (*Nonce-List*) and revokes each nonce whenever its used first time.

Nonces can be updated by hand-shaking, after successful key initiation phase. Both *NS* and the end-device might agree on a new nonce and that nonce might be written to a *non-volatile memory* of the end-device. The old nonces would be moved to the "used nonces list" and the new nonce should be moved to "valid nonces list".

By this way, any attempt to use an old nonce would not cause any security breach in the network. Besides, these attempts can easily be detected by the *NS* by comparing the nonce with the nonce lists. The join request can be granted if only the submitted nonce is in the *valid nonces list* otherwise, if the nonce is in the *used nonces list*, then this is a possible situation of cloning, node-capture or replay attack!

*6.3. Introduction of End-to-End Encryption between Servers*

This topic has prime importance, as it will completely prevent any kind of man-in-the-middle attacks towards servers. Although it is mentioned in the LoRaWAN v1.1 specification [16] that the communication in between the servers is considered as secure.

*6.4. Inclusion of Padding*

To divert packet-sniffing attacks, it is always a good practice and counter-measure to set packet length to a certain constant value and apply padding when needed. Eventually, this will come with a price of increased airtime of the added extra padding data.

*6.5. Packet Count Limitation*

This opens a venue for attacks for LoRaWAN, since the communication per *ED* is limited and under quota. For instance, each *ED* is allowed to transmit at most 14 packets per day (maximum packet payload of 12 Bytes), including the acknowledgments for confirmed up-links [38].

*6.6. Time Synchronization*

Hence time synchronization of the *ED*s is provided within v1.1 (*DeviceTimeReq/Ans*), this can be efficiently used as a *time-stamp* (with 1–2 s precision) to improve the freshness and integrity of the messages.

*6.7. Gateway Related Improvement*

An authentication mechanism for the *GW*s is necessary in order to prevent the network from *Rogue-Gateway* attacks. For instance, mutual authentication can be performed between the couples of *ED-GW* and *GW-NS*.

*6.8. Server Trust Related Suggestion*

In LoRaWAN, servers need to be trusted entities otherwise, they can create single point of failure for the network. By default, all the servers in LoRaWAN are assumed to be trusted entities as mentioned in the specification [16].

*6.9. End-Device Related Improvement*

Physical capture threat against end-devices (and eventually extraction of security parameters) can be neutralized by usage of tamper-resistant hardware for the storage of the keying materials in the end-devices. Although this comes with an extra cost in safety-critical installations where cyber threats need to be mitigated, the use of tamper-resistant hardware should be mandatory.

## 7. Conclusions and Future Work

LoRaWAN is one of the most comprehensive and adopted LPWAN technologies, which is rapidly growing in IoT applications, especially in smart meters and oil and gas operations. LoRaWAN v1.0 had security breaches and drawbacks, which were addressed by the release of LoRaWAN v1.1 recently. Although it has improved security properties compared to the previous version, LoRaWAN v1.1 still has some security risks, some introduced by the new security framework, others by not being covered by the specification, which warrants attention from developers. In this article, the authors' aim was to shed additional light to these risks by presenting a comprehensive Security Risk Analysis, along with providing some remedies such as prevention and mitigation strategies.

In this work, we have analyzed the security risks of LoRaWAN v1.1 by using ETSI guidelines and created a threat catalog for LoRaWAN v1.1 along with discussions and analysis in view of scale, impact and likelihood of each threat. We also discussed several aspects that are more related to practical implementation of the technology and while they are not necessarily part of the spec, they are important to highlight for practitioners. Furthermore, we also provided our insights regarding security improvements of LoRaWAN that might be adopted for up-coming versions.

According to the result of this security risk analysis, LoRaWAN v1.1 appeared to have a couple of relevant security threats (especially vulnerabilities against end-device physical capture, rogue gateway and replay attacks). Other than the security flaws mentioned in this text, LoRaWAN v1.1 has proven to be more secure and reliable, compared to its earlier version (v1.0). Therefore, with the addition of these new features of roaming and mobility support (the existing ones were: long range, wide availability of low-cost devices, high community support, easily implementable devices), we predict LoRaWAN to be more and more attractive for IoT applications.

In the future, we are planning to do a comprehensive security analysis of LoRaWAN v1.1 with one of the security proof tools, such as Tamarin [40] or Scyther [41]. We will also propose new techniques and/or methodologies to tackle these newly recognized security problems. Finally, our future work will focus on finding secure key distribution mechanisms that can be used by LoRaWAN v1.1, in the distribution phase of root keys (i.e., the AES-128bits root keys: *AppKey* and *NwkKey*).

**Author Contributions:** I.B. and N.P. conducted the research along with analysis and results. The manuscript is also written by the same authors. M.G. has provided the funding and supervision of this research.

**Funding:** This work (research as well as open access publication) was supported partially by grants: 20201010 (SMART Project) of the European Regional Fund (ERUF), 20150367 (TIMELINESS Project) of the Swedish Knowledge Foundation (KKS), and the Portuguese funding institution FCT—Fundação para a Ciência e a Tecnologia, under the sabbatical leave fellowship SFRH/BSAB/128459/2017.

**Conflicts of Interest:** The authors declare no conflict of interest.

**Abbreviations**

The following abbreviations are used in this manuscript:

| | |
|---|---|
| ABP | Activation by Personalization |
| ADHE | Authenticated Diffie-Hellman key Exchange algorithm |
| ADR | Adaptive Data Rate |
| AFCntDwn | Application Frame Counter Down |
| AppKey | Application Key, root key for *AppSKey* |
| AS | Application Server |
| CSS | Chirp Spread Spectrum |
| DER | Data Extraction Rate |
| DevAddr | Device Address of end-device |
| DevNonce | Device Nonce of end-device |
| DoS | Denial of Service |
| ED | End Device |
| ETSI | European Telecommunications Standards Institute |
| FCntUp | Frame Counter Up |
| fNs | forwarding Network server |
| FNwkSIntKey | Forwarding Network Session Integrity Key |
| GW | Gateway |
| HSM | Hardware Security Module |
| IoT | Internet of things |
| JoinNonce | Nonce of Joining server |
| JS | Joining Server |
| JSEncKey | Joining Session Encryption Key |
| JSIntKey | Joining Session Initiation Key |
| LoRa | Long-Range LPWAN technology |
| LoRaWAN | An application of LoRa |
| LPWAN | Low Power Wide Area Network |
| MIC | Message Integrity Check |
| MITM | Man-In-The-Middle attack |
| NFCntDwn | Network Frame Counter Down |
| NS | Network Server |
| NwkKey | Network Key, root key for session keys: *FNwkSIntKey*, *NwkSEncKey*, *SNwkSIntKey* |
| NwkSEncKey | Network Session Encryption Key |
| OTAA | Over the Air Activation |
| PKI | Public Key Infrastructure |
| RD | Rogue Device |
| RF | Radio Frequency |
| SE | Secure Elements |
| SNwkSIntKey | Serving Network Session Integrity Key |
| SWOT | Strength, Weakness, Opportunity and Threat analysis |

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
