# Peer review of "Security Risk Analysis of LoRaWAN and Future Directions"

_futureinternet, doi:10.3390/fi11010003_

Round 1

Reviewer 1 Report

The paper “Demystifying the Security of LoRaWAN v1.1” presents a security analysis of LoRa technology.

I recommend the authors to change the title of the paper to a more suggestive one.

However, the scientific background and rationale for certain design decisions are argued very weak.

Figure 1 must be reformatted because is of poor quality.

Also, the entire structure of the paper must be reformatted. There are to many subsections with information from LoRa specification.

Although the addressed research area is of interest, the paper is not clear on the innovation it brings. This might be due to a poor organization and description of the developed work. Moreover, few references have been made to the existing state of the art research. The authors must state the main contribution of the paper.

The theoretical study must be completed by some simulations or practical tests.

Author Response

Please refer to the attached Response Letter.

Reviewer 2 Report

In this paper, the authors discuss the security vulnerabilities of LoRaWAN v1.1. The paper is an extension of a paper published previously in an ACM workshop. Security in LoRaWAN is an important topic and the authors presented a good analysis of LoRaWAN main vulnerabilities and some possible solutions. The paper does not come up with great contributions to the state of the art, but can be useful for readers who are interested in learning about LoRaWAN security.  

There are a few minor corrections to be made:

- lines 56 and 57 on page 2: two sentences in a row are beginning with "therefore", making it repetitive. 

- Figure 1: this figure is not needed, because these details about layers do not fit well in a paper introduction. Besides, the figure was extracted from other paper, meaning it is not even an original contribution from the authors.

- Line 115 on page 3: "random generator": instead of using quotes to denote a random generator which does not work properly as so, it is better to explain the situation. This use of quotes may confuse some readers.  

- Section 4.1.3 on page 8: the subsection has only one paragraph with two lines. Is it really necessary to break the subsection 4.1 into three subsections? The same situation can be observed in other parts of the paper. 

Author Response

(The authors gave the same response as above.)

Round 2

Reviewer 1 Report

I recommend the paper to be accepted.

Author Response

Please see attached response letter.
